# Diagnosis by Endoscopic Ultrasonography-Guided Sampling through the Lower Gastrointestinal Tract

**DOI:** 10.3390/diagnostics14010064

**Published:** 2023-12-27

**Authors:** Jingyuan Wang, Yue Liu, Chang Wu, Jiayu Fan, Zhendong Jin, Kaixuan Wang

**Affiliations:** Department of Gastroenterology, Changhai Hospital, Naval Medical University, Shanghai 200433, China; wjy706815@163.com (J.W.); liuyue200201@163.com (Y.L.); speedywc@163.com (C.W.); fanjiayu07@163.com (J.F.)

**Keywords:** endoscopic ultrasound, EUS-guided fine-needle aspiration, EUS-guided fine-needle biopsy, lower gastrointestinal tract lesions, pelvic lesions, diagnosis

## Abstract

Endoscopic ultrasound-guided fine-needle aspiration/biopsy (EUS-FNA/FNB) is very safe and has a high diagnostic rate for upper gastrointestinal lesions, especially pancreatic lesions, but its application in the lower gastrointestinal tract has rarely been reported. Due to the tortuous course of the colorectum, with the sigmoid colon particularly prone to perforation, most endoscopists are reluctant to perform lateral-sector endoscopic ultrasound scanning without a water-bag protection for the puncture. The ultrasonic endoscopy and flexible puncture needle techniques recently introduced into clinical practice have made ultrasound-guided puncture safer and more convenient. In addition, endoscopists have carefully tested various protective measures to improve the safety of the lower gastrointestinal puncture, substantially increasing its clinical feasibility. In this article, we review the iterations of endoscopic ultrasound equipment introduced in recent years and the many ingenious ideas proposed by endoscopists regarding lower gastrointestinal puncture.

## 1. Introduction

Endoscopic ultrasonography (EUS) is a combination of high-frequency ultrasound and endoscopy, which, in addition to endoscopic vision, can provide real-time images of intra-abdominal lesions through the lumen of the digestive tract, shorten the distance between the ultrasound probe and the lesion, and avoid the influence of abdominal wall attenuation and gastrointestinal gas. It offers a higher resolution for certain lesions than CT and MRI, filling the gaps left by general endoscopy, surface ultrasound, and sectional tomography imaging. As a relatively mature technique, endoscopic ultrasound-guided fine-needle aspiration/biopsy (EUS-FNA/B) can be used for the diagnosis and treatment of not only digestive tract diseases, but also lesions in adjacent organs within the abdominal cavity [1]. In most cases, EUS-FNA/B is performed through the upper GI tract. The European Society of Gastrointestinal Endoscopy (ESGE)’s clinical guidelines specify that endoscopic ultrasound (EUS)-guided sampling is indicated for subepithelial lesions; diffuse esophageal/gastric wall thickening; solid and cystic pancreatic masses; unexplained biliary strictures; mediastinal lesions not associated with lung or esophageal cancer; esophageal, gastric, and rectal cancers; lymph nodes of an unknown origin; adrenal masses; and focal liver lesions [2,3]. However, some lower GI and pelvic lesions that are difficult to access through the upper gut wall may be sampled through the lower GI wall. Deep subepithelial lesions often occur, as well as extramural compression in some cases. For these lesions, routine bite-on-bite and mucosal-incision-assisted biopsies of these lesions only indicate normal colonic mucosa. Therefore, a biopsy reaching deep inside the puncture is very important [4,5]. Moreover, EUS-guided puncture drainage has gradually become a safe and effective treatment method for some pelvic abscesses that are difficult to drain percutaneously [6,7,8].

## 2. Clinical Applications

EUS can be used to study the relationship between the gastrointestinal tract wall and tumor location and can accurately distinguish extramural compression and intramural tumors. For subepithelial lesions, EUS has significant diagnostic advantages compared to contrast-enhanced cross-sectional imaging. At the same time, the guidelines do not recommend conducting bite-on-bite biopsies in the evaluation of SELs before EUS [9]. EUS can diagnose duplication cysts, lipomas, varicose veins, and ectopic pancreas. A lipoma appears as slightly yellow under white-light endoscopy and shows a “pillow sign”, i.e., pressing with closed biopsy forceps easily deforms the lipoma [10]. However, EUS alone cannot easily differentiate and diagnose other types of solid subepithelial lesions (SELs) such as neuroendocrine tumors, leiomyomas, granulosa cell tumors, gastrointestinal stromal tumors, schwannomas, and malignant metastases, especially for fourth-layer tumors or smaller SELs, which are more prone to a misdiagnosis [11,12]. Although contrast-enhanced harmonic EUS and ultrasound elastography can enhance the characterization of SELs, they cannot replace EUS-guided sampling [13,14]. Therefore, many lesions are difficult to classify through ultrasound images alone, and a routine biopsy often cannot reach the structures below the mucosa, resulting in an unstable diagnostic rate with the bite-on-bite technique for biopsy samples of SELs located in the third or fourth layers of the endoscopic ultrasound (EUS), hovering between 17% and 94% [15,16]. EUS-guided puncture sampling is a feasible approach, with the advantage of a lower bleeding risk compared to the jumbo unroofing technique [17]. Although a previous study confirmed that mucosal incision-assisted biopsy is superior to ultrasound-guided fine-needle aspiration biopsy in diagnosing small lesions in the upper and lower layers of the stomach [18], EUS-guided aspiration biopsy may still be preferred in the colon, especially when space-occupying lesions make mucosal resection difficult. Moreover, the pathological results for the EUS-guided sampling and endoscopic resection of lesions are highly consistent, and fewer complications are associated with EUS-guided sampling [19,20]. When diagnosing subepithelial lesions, EUS-FNB often requires fewer passes than EUS-FNA [21].

### 2.1. Transrectal Sampling

The 2017 ESGE guidelines only mention transrectal sampling. Perirectal lymph node puncture is not recommended for the staging of patients with first-episode rectal cancer. However, if a perirectal mass is found during the monitoring period after rectal cancer, it should be punctured to check for distant metastasis (the common metastatic location is the extramesenteric lymph nodes) because this may expand the treatment area of surgery or radiotherapy [2]. The puncture diagnosis of cancer foci that have metastasized to the rectal wall has also been reported. For example, Ferga et al. recorded six cases of rectal wall metastases, whose morphological characteristics under endoscopic ultrasonography usually mimic those of linitis plastica, i.e., diffuse, circumferential, hypoechoic wall thickening [22]. In this case, puncturing can identify the source of malignant cells. In addition, various pelvic lesions and subepithelial lesions in the intestinal wall, such as sarcomas, melanomas, gastrointestinal stromal tumors, fibromas, leiomyomas, lipomas, schwannomas, neuroendocrine tumors, pelvic cysts/abscesses, and postoperative adhesion, can be pathologically diagnosed by rectal puncture [5,23,24].

### 2.2. Sampling via Sigmoid Colon

Nakano et al. performed a routine endoscopic biopsy on a patient with sigmoid colon wall thickening and severe stenosis. Because the endoscope could not pass through the stenotic segment, the biopsy could only be conducted around the stenosis. Although six samples were taken, no clear pathological diagnosis could be obtained [25]. Kishimoto et al. used a safer ropeway method of EUS-FNA supported by fluoroscopic guidance to complete the puncture in a case of sigmoid-endometriosis-induced intestinal stenosis. The ropeway method is the single-guidewire-assisted method mentioned above. The article also mentioned another difficulty of trans-sigmoid puncture: the sigmoid colon is not directly attached to the posterior abdominal wall, increasing its mobility. This hinders the accurate insertion of the needle, and excessive penetration may cause endometrial cells to be sown into the abdominal cavity. This problem complicates EUS-FNA/FNB for lesions occupying a small intramural space, such as small GISTs. Hoda et al. reported that the diagnostic rate of gastrointestinal stromal tumors less than 10 mm in diameter by EUS-FNA is 40–50%, which is a very low level. Thus, only attempting to puncture lesions that are at least 10 mm thick may be safer. In addition, we can reasonably speculate that puncturing through the ascending and descending colon is much more secure because they are “fixed” to the posterior abdominal wall [26]. Other case reports include the diagnosis of a pancreatic ductal adenocarcinoma by the fine-needle biopsy of an anatomically altered pancreas through the sigmoid colon with fluoroscopic assistance [27], the diagnosis of post-transplant lymphoproliferative disease by the fine-needle biopsy of a pancreas allograft through the sigmoid colon [28], and the diagnosis of the pelvic recurrence of an anal squamous cell carcinoma through the sigmoid colon [29].

### 2.3. Sampling via Descending, Transverse, and Ascending Colon

Akahoshi et al. used the single-cannula-assisted technique mentioned above for the fine-needle aspiration of a descending colon schwannoma [4]. Trikudanathan et al. reported a case of a schwannoma at the splenic flexure of the colon [30]. Schwannomas are hard in texture and often originate from the third and fourth layers of an ultrasound (muscularis propria) [31]. A conventional colonoscope biopsy often only shows lymphocyte aggregation, so ultrasound-guided puncture is particularly important for such masses. Fehring et al. reported a case of a recurrent follicular lymphoma in the transverse colon diagnosed by EUS-FNA. A lymphoma presents as a deep-seated space-occupying lesion outside or inside the colorectal wall, often causing intestinal obstruction, which is difficult to identify by colonoscopy and mucosal biopsy. EUS-guided sampling can determine the “hidden” etiology of this intestinal obstruction [32].

Many lesions are difficult to pathologically diagnose using conventional endoscopic mucosal biopsy due to the following factors: (1) compression causes intestinal stricture, so ordinary colonoscopy can only obtain marginal samples from beneath the lesion, reducing the diagnostic yield. (2) If the lesion is deeper within or even outside of the intestinal wall, the biopsy results often indicate normal mucosa or reactive inflammation. In this case, puncture biopsy can penetrate deep into the lesion to obtain satisfactory histological specimens. However, as mentioned above, the higher mobility of the sigmoid colon hinders its puncture, and the higher mobility of the transverse colon may cause similar problems. The size, location, and texture of the target lesion; its relationship with surrounding tissues; and the experience of the endoscopist affect the difficulty of a puncture. Most relevant publications are case reports. We look forward to further case–control studies considering the location of the intestinal segment.

## 3. Development of New Ultrasonic Endoscope

Due to the multiple physiologic curvatures in the sigmoid colon, advancing a linear-array ultrasound endoscope relying solely on endoscopic vision and ultrasound images is dangerous and prone to perforation. Reviewing all available cross-sectional images of the pelvis is critical to evaluating the left colon and excluding large diverticula before advancing a linear EUS scope [30]. Subsequently, a suitable echoendoscope is chosen or auxiliary equipment is used to complete the puncture.

(1) Longitudinal-axis ultrasound endoscope: When the scanning direction of the ultrasound endoscope during puncture is parallel to the long axis of the endoscope, this is known as a longitudinal-axis ultrasound endoscope. During puncture, the needle is constantly monitored by ultrasonic imaging so that the puncture operation is accurate and safe. Most ultrasound endoscopes used for puncture are electronic linear-array scanning ultrasound endoscopes. The latest endoscopic ultrasound host machine has a series of new functions, including the tissue harmonic (THE), elastography (ELST), pulsed-wave Doppler (PW), high-definition blood flow (H-flow), and contrast harmonic EUS (CH-EUS) modes [33,34,35]. CH-EUS can be used to indicate pathological staging and improve the differentiation of T1 and T2 lesions in cancer nodules [17]. This new machine can perform image quality adjustment and frequency switching, sensitivity time control (STC), 16-segment gain, 8-segment contrast, observation range adjustment, image direction selection, 64-segment image rotation, endoscopic and ultrasound image selection display or simultaneous display (“picture in picture” display), and target lesion size or volume measurement. New EUS hosts such as the Fujifilm su-7000 and su-8000 systems also introduced functions such as synchronous dual-plane reconstruction (DPR), a built-in trackball, image memory playback, and a color–power Doppler function, which improved the safety of EUS-guided puncture.

The vast majority of endoscopic ultrasound applications for puncture sampling employ the anterior oblique endoscopic view (where the angle between the direction of view and the axis of the endoscopic body is 40~55°) and lateral ultrasound scanning (with a scanning range of 120~180°). Another technique is to use the anterior direct endoscopic view and anterior oblique ultrasound scanning (scanning range of 90°), that is, a forward-viewing linear-array echo endoscope (FV-EUS). The first generation of FV-EUS included the XGIF-UCT160 (Olympus Medical Systems, Tokyo, Japan), and the new generation is represented by the TGF-UC260J (Olympus Medical Systems, Tokyo, Japan). Nguyen-Tang et al. used FV-EUS to puncture right colon and cecum subepithelial lesions, achieving a high technical success rate (5/6), with only one case failing due to lymph node calcification [26,36].

Conventional linear-array ultrasound endoscopes are less flexible, especially for right colon lesions. New curved linear echo endoscopes (such as the eg-580ut, Fujifilm, Tokyo, Japan, and eg34-j10u, Pentax, Tokyo, Japan) have flexible tips. Compared with traditional curved echo endoscopes, they can be more safely inserted into the right colon. Additionally, compared with forward-viewing linear-array echo endoscopes, they have a wider ultrasonic scanning range and a thinner endoscopic body [37].

(2) Micro ultrasonic probe: Most micro-probes have a length of over 2 m and an outer diameter below 2.6 mm at the insertion site, making them suitable for the vast majority of colonoscopes with a length of approximately 130 cm and a working channel diameter of 2.8 mm. A micro ultrasonic probe can easily pass through the narrow part of a lesion that is inaccessible to an endoscope, which is very helpful for diagnosing the depth of invasion of advanced cancer and extramural lesions. High-frequency micro-probes can reveal the structure of the intestinal wall with a high resolution, which is very useful for diagnosing small, flat lesions, such as early cancer or other superficial intramural lesions. Micro ultrasonic probes most often employ circular scanning, and they have not yet been used to guide a puncture [38,39].

## 4. Development of Puncture Needles

FNB needles have side slots or cutting tips with special geometric structures that capture the core tissue, making them more effective for cutting tissues than FNA needles. FNB needles can more easily obtain specimens with a complete tissue structure. Based on the results of previous RCTs and meta-analyses, FNB needles have several advantages over FNA needles: (1) they require fewer passes to obtain enough tissue for histological diagnosis; (2) they provide higher histological quality [40]. The first-generation FNB reverse-bevel needle failed to significantly improve upon the results of FNA needles. In recent years, new FNB needles with altered cutting tips have achieved a higher diagnostic efficiency. A network meta-analysis by Gkolfakis et al. showed that Franseen needles and fork-tip needles are significantly superior to reverse-bevel needles and FNA needles in terms of diagnostic accuracy and specimen adequacy [40].

Currently, 19G (outer diameter 1.07 mm, inner diameter 0.69 mm), 20G (outer diameter 0.91 mm, inner diameter 0.58 mm), 22G (outer diameter 0.71 mm, inner diameter 0.41 mm), and 25G (outer diameter 0.51 mm, inner diameter 0.25 mm) FNA/B needles are available on the market. The 19G needle is thicker and stiffer than other needles, making it more difficult to manipulate in angled locations such as the duodenum. In addition, after the needle tip direction is changed by the forceps lifting device, the stainless-steel needle cannot easily return to its original shape, preventing secondary needle sampling. The use of shape-memory metals solved the above problems. Benchtop tests conducted by Itoi et al. proved that, in contrast to traditional 19G needles, 19G nitinol needles can be inserted at the optimal angle with minimal resistance under various endoscope bending conditions [41]. A multicenter, prospective, randomized, double-blind study of EUS-FNB needles applied to solid pancreatic masses by Hann et al. showed no significant difference in diagnostic rates between 19G stainless-steel needles (65.9%) and 19G nitinol needles (68.3%) (*p* = 0.73) [42]. However, nitinol needles reduce the puncture time and technical failure rate (i.e., the needle cannot be bent or inserted twice). The nitinol needles on the market have gradually developed through the 19G, 22G, and 25G models, which can enter difficult-to-reach lesions, such as those situated in the uncinate process of the pancreas, without affecting the position of the lens. This mechanical property is required when puncturing a proximal colonic target lesion in the lower gastrointestinal tract. Akahoshi et al. used a nitinol needle to puncture submucosal tumors in the descending colon [4]. The commonly used FNB needles are shown in Table 1.

## 5. Sampling Process

### 5.1. EUS-FNA/FNB Standard Sampling Process

The TCB (Trucut biopsy) needle has almost been completely abandoned due to its limitations, including the serious puncture damage caused by the large outer diameter, hard needle body, and poor handling performance. Endoscopists comprehensively consider the anatomic location and type of lesion, determine the processing mode according to the specimen and their own experience, and select the appropriate type of puncture needle. The needle is deployed to the appropriate position under the guidance of endoscopy and side-view ultrasound probes, with or without the injection of an acoustic contrast agent, and the needle stylet is removed after the needle is inserted into the lesion. The operator connects a 10 mL or 20 mL negative-pressure syringe to the needle to apply negative pressure and repeatedly enters and exits the lesion in a fan-shaped pattern (if the lesion is cystic, the first needle should be used to suck out the cystic fluid). Several articles have mentioned the methods of applying micro-negative pressure or no negative pressure at all [43]. The choice of negative pressure has advantages and disadvantages. A high-negative-pressure puncture can provide a substantial amount of tissue but involves a large area of blood contamination, while a low-negative-pressure puncture can provide a small amount of tissue with a small area of blood pollution. Some physicians also recommend “improved wet suction” [44]. No definitive conclusion has been drawn as to the impact of the suction method on diagnostic accuracy. Different numbers of needles are inserted based on the type of lesion, rapid on-site evaluation (ROSE) by a cell pathologist, or macroscopic on-site evaluation (MOSE) by an endoscopist. To prevent the destruction of normal tissue by residual negative-pressure suction, the syringe stopcock is separated from the needle hole before the needle is removed from the target lesion. The obtained cell clusters and tissue strips are subjected to cytological smears and histological tests, and additional genetic testing may be performed if necessary [1].

### 5.2. EUS-FNA/FNB Sampling Process through the Lower Gastrointestinal Tract

(1) Forward-viewing linear-array echo endoscopy (FV-EUS) + fluoroscopy-guided sampling: Novel FV-EUS has unique advantages, such as a forward view instead of a side view (most conventional longitudinal-axis endoscopic ultrasound equipment have a 40~50° anterior squint view), a straight channel port, a wider bending angle, and a shorter all-inclusive distal end. These features provide a stronger puncture force and superior access to the proximal colon, reducing the risk of perforation. However, this technique has the following shortcomings: (1) a larger endoscopic diameter and (2) a narrower ultrasound scanning range (90°) compared with traditional CLA-EUS (180°). These drawbacks increase the difficulty of scanning and operation. The solution is to incorporate fluoroscopy-guided sampling. The C-arm X-ray machine is used to change the plane of the X-ray film to compare the tip position of the endoscope under fluoroscopy and the position of the target lesion according to cross-sectional imaging (such as CT, MRI, or PET/CT). Then, the target lesion is identified using the ultrasonic visual field, and the puncture is performed. Fluoroscopy helps to control the curvature of the ultrasound endoscope when reaching the proximal colon and to locate tumors outside the colon in the narrow scanning area of the ultrasound endoscope. A retrospective study by Thinrungroj et al. showed that technique success (100%), a shorter procedural time (median of 31 min), a higher diagnostic accuracy (92%), and a lower adverse event rate (0) were guaranteed when performing needle sampling with FV-EUS under fluoroscopy [45].

(2) Overtube/guidewire technology:

① Using an overtube alone. If the lesion is located in the descending colon, a conventional colonoscope is first inserted into the oral side of the lesion, and then a marker clip is placed on the side of the lesion. The endoscope is straightened to straighten the sigmoid colon, and an overtube is inserted across the sigmoid colon. Next, the colonoscope is removed, and a linear-array scanning ultrasonic endoscope is inserted along the overtube. Finally, a puncture sample is collected [4].

② Using a guidewire alone. Similar to using an overtube alone, the guidewire is first deployed through the colonoscope over the stenosis of the lesion, and then the endoscopic ultrasound is deployed along the guidewire [46].

③ Using both a guidewire and an overtube. If the lesion is located in the right colon, the colonoscope is inserted into the cecum first. In this procedure, the sigmoid colon is straightened after sufficient ring shortening; subsequently, a hard guidewire is inserted through the colonoscopic working channel into the cecum and coiled there. Then, an overtube is inserted across the sigmoid colon and carefully exits the colonoscope. Finally, the EUS equipment is inserted via the overtube and the guidewire [47]. The operation flow of this technology is shown in Figure 1.

If the right colonic lesion is sampled using the novel flexible curved echoscope described above, only a guidewire is required for assistance [37].

(3) Water injection technique: An endoscopic physician uses a standard colonoscope to fill the intestine with clean water or normal saline to prepare for the water injection technique. Under the guidance of ultrasound imaging, the echoendoscope is introduced into the splenic curvature and slowly rotated until the lesion is seen in the ultrasound field for puncture. The water-filling technique, in which the water acts as an ultrasonic glue to enhance the sensitivity of the ultrasound, is much simpler. The water also lubricates and fills the intestine, making it easier for the EUS device to pass through the curved sigmoid colon, while acting as a “protective layer” to reduce the risk of perforation. Mangiavillano et al. expect the water injection technique to be applied to the EUS of the transverse or right colon, but this must be explored in further studies [48].

## 6. Complications

EUS-FNA/FNB is very safe, and complications such as bleeding, infection, gastrointestinal perforation, and needle implant metastasis are extremely rare [1,48]. The rapid development of new technologies mentioned above has reduced the risk of significant concerns such as bleeding and perforation during lower gastrointestinal EUS-FNA/FNB. Other major concerns are pain, infection, and further complications. Unlike the upper GI, the lower GI is full of bacteria. Levy et al. published two articles in 2007 and 2014 on the post-procedural complications of EUS-FNA in the lower gastrointestinal tract.

A prospective study on bacteremia and the complications of EUS-FNA for rectal and perirectal lesions by Levy et al. showed that six out of one hundred patients undergoing EUS-FNA produced positive blood culture results before or after puncture. Among them, four cases were believed to be contaminated during the culture process, and two cases were believed to be bacteremia: one case of Bacteroides fragilis and one case of Gemella morbillorum. The sample that tested positive for Bacteroides fragilis originated only from the blood culture before puncture, so it was not caused by FNA. None of these 6 patients showed any signs or symptoms of infection. The authors suggested that the EUS-FNA of the lower gastrointestinal tract should be considered as a low-risk procedure for infectious complications, without the need for the prophylactic use of antibiotics [49].

Levy et al. conducted a prospective evaluation of adverse events after the EUS-FNA of the lower gastrointestinal tract in 2014. Due to the early publication date, the main surgical site was the rectum, and old Trucut needles were used when conducting the biopsy. As mentioned above, Trucut needles are a type of needle with a higher risk of adverse events. After excluding the influence of other operations on the same day, the authors suggested that EUS-FNA was still associated with a higher incidence of complications, especially serious grades 3–4 adverse events. However, this conclusion is controversial: for example, pain is often caused by malignant tumor itself, and patients with post-procedural pain often experience pre-procedural pain; the only factor associated with new or increased bleeding was hot snare polypectomy, while EUS-FNA itself did not increase the risk of bleeding. Among 502 patients, only 5 (1%) presented a grade 1 fever according to the Common Terminology Criteria for Adverse Events (CTCAE) The incidence of perforation was very low (1 case/502 cases), and none exhibited a serious infection such as an abscess. The incidence of perforation and appendicitis was very low (1/502 each). Furthermore, it is worth noting that no significant difference in risk was found between patients who received prophylactic antibiotics and those who did not. Therefore, the higher post-procedural AE rate did not seem to have been caused by EUS-FNA, and the incidence of infection was very low [50].

Cazacu et al. reviewed their 10-year working experience with the EUS-guided sampling of the lower gastrointestinal tract. Twenty-one patients underwent EUS-guided tissue collection, and only one surgery-related adverse event was reported. They argued that EUS-guided tissue acquisition is a safe operation. All patients treated with EUS-FNA/FNB at their institution received perioperative antibiotics. However, the sample size of this retrospective report was small, and no control group was included [5].

Although the recommendations are tentative, the guidelines do not require the prophylactic use of antibiotics for the puncture of either upper or lower gastrointestinal solid lesions. Generally speaking, although less relevant literature exists in this regard, EUS-FNA/FNB in the lower gastrointestinal tract has a low risk of infection. No convincing conclusion has been reached as to the prophylactic use of antibiotics. Further clinical research is needed, especially studies based on upgraded puncture needles and echo endoscopy. For both the upper and lower gastrointestinal tracts, if the target lesion is cystic, fluoroquinolones or β-Lactam antibiotics should be used prophylactically [51,52].

## 7. Summary and Prospects

EUS-guided sampling is a mature and reliable method for obtaining pathological results regarding gastrointestinal and peripheral lesions. Some colonic wall lesions are located in the muscular or even serosal layer below the mucosa, and negative results are often obtained through conventional colonoscopy and biopsy forceps. In addition, some abdominal lesions are located deep in the pelvic cavity, making the puncture of the abdominal wall or upper gastrointestinal tract difficult. Clear pathological results may determine the treatment plan, so it is necessary to obtain samples by puncturing the lower gastrointestinal tract.

With the emergence of safer and more flexible ultrasound endoscopes, new FNB needles made from shape-memory metal, and the use of wires or overtubes to guide punctures (creatively proposed by endoscopic physicians), EUS-guided sampling has gradually extended its application range from the rectum to the proximal colon. The abundance of bacteria in the colon and rectum leads most physicians to apply antibiotics prophylactically. Due to the limited number of lower gastrointestinal puncture cases in diagnostic and treatment centers, we lack authoritative research results as to the incidence of infection in lower gastrointestinal EUS-FNA/FNB puncture. Therefore, the recommendations for the use of antibiotics to prevent infection before or after surgery cannot be replicated from the upper gastrointestinal puncture scenario, and further RCTs are needed in this regard.

Ultrasound-guided lesion puncture through the lower gastrointestinal tract is a beneficial supplement to percutaneous puncture and upper gastrointestinal puncture. The existing reports demonstrate the effectiveness and safety of this technology.

## Figures and Tables

**Figure 1 diagnostics-14-00064-f001:**
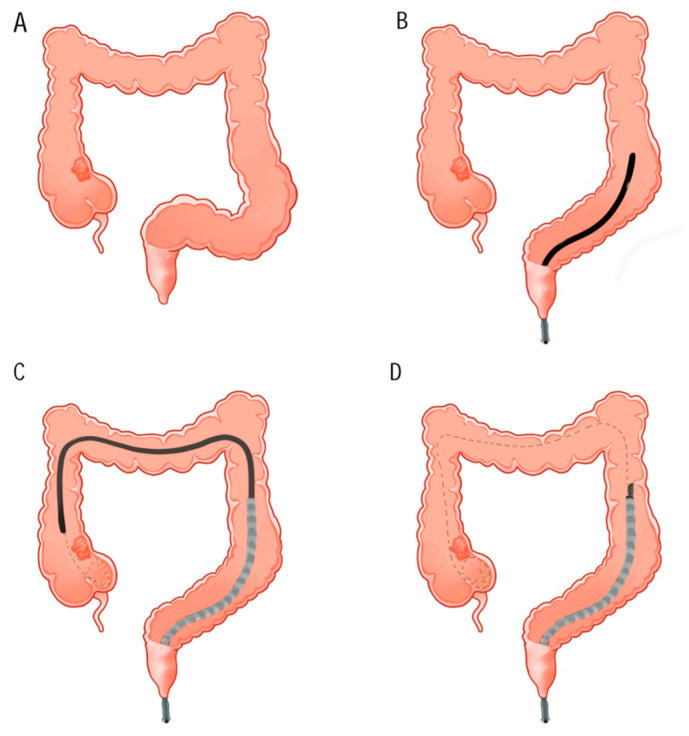
Puncture of lesions in the ascending colon assisted by guidewire and overtube. (**A**)—The lesion is located in the initial segment of the ascending colon, and the sigmoid colon is in a relaxed and curved state. (**B**)—During colonoscopy, the overtube is preloaded onto the colonoscope, and the sigmoid colon is straightened after the loops are fully compressed. (**C**)—The guide wire is inserted into the cecum through the working channel of the colonoscope and coiled at the distal end of the lesion; then, the overtube is inserted along the colonoscope. (**D**)—After the withdrawal of the colonoscope, the EUS device is anastomosed with the guidewire through the previously loaded biliary sphincterotomy device and advanced to a location near the lesion with the dual assistance of the guidewire and overtube.

**Table 1 diagnostics-14-00064-t001:** Characteristics of commonly used FNB needles.

Trade Name	Producer	Shape Characteristics	Models
Acquire^®^/Acquire^®^S	Boston Scientific, Marlborough, MA, USA	Known as the “Franseen” needle; shaped like a crown, comprising three symmetrical needle points with fully formed cutting heels.	22G, 25G stainless-steel needles; 19G nitinol needle
SharkCore^®^	Medtronic, Dublin, Ireland	Known as the “fork-tip” needle; resembles a shark’s head, with a total of four needle tips: one long, one short, and two extremely short, which together form six distal cutting-edge surfaces.	22G, 25G stainless-steel needles; 19G nitinol needle
EchoTip ProCore^®^	Cook Medical, Bloomington, IN, USA	The front end of the needle tip is similar to an FNA needle with side fenestration, including the first generation of the “reverse inclined plane” needle and the new generation of the “forward inclined plane” FNB needle.	19G, 20G, 22G, 25G stainless-steel needles
EZ Shot 3plus^®^	Olympus, Shinjuku, Tokyo, Japan	Known as the “Menghini” needle; no unique structural characteristics in structure; some institutions in Europe and the United States believe that it cannot be classified as an FNB needle.	19G, 22G, 25G stainless-steel needles; 19G, 22G nitinol needles

## Data Availability

Not applicable.

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
