# Peer review of "Diagnosis by Endoscopic Ultrasonography-Guided Sampling through the Lower Gastrointestinal Tract"

_diagnostics, 2023, doi:10.3390/diagnostics14010064_

Round 1

Reviewer 1 Report

Comments and Suggestions for Authors

The subject of the article is interesting and useful. Unfortunately  it is written in poor English.

Some abbreviations are not clarified (TCB in line 53).

The chronology of the article can be changed: first the indications to the examination, than the echoendoscopes and technique of examination than biopsy and needles finally complications.

The difference between FNA and FNB and the indications to choose either FNA or FNB are not sufficiently explained.

The description of echoendoscope types and needle types without the figures showing the differences is difficult to get through. The word “strabismus” should be rather replaced with ”oblique view”. What does ”mirror body” mean (line 150)?

Are there micro ultrasonic probes manufactured only for the use with gastroscopes or are they also compatible with colonoscopes?

Lines 170-175 – What is the advantage of X-ray?  Does it serve for control of echoendoscope flexures while reaching the proximal colon or does it rather help to find the tumors outside the colon in narrow scanning field of echoendoscope placed in distal colon?

Are there any data about introduction forward-viewing echoendoscope with the help of the guidewire inserted via colonoscope or such help in this type of echoendoscope is not needed?

Water injection technique- is this technique basing on ”follow the water” rule or only helps to make flexures milder? If the first is true – is it rather endoscopic or ultrasonic view more useful to guide the way?

Line 270: SEL are subepithelial, not subcutaneous lesions.

Comments on the Quality of English Language

line 66 - contamination not pollution

line 81  passes not needles

negative pressure - suction

line 117 - word redundant should be replaced

line 173 - procedural not surgical

Author Response

Thank you for your correction!
One advantage of forward-looking ultrasound endoscopy is that it is less prone to perforation. Currently, in existing reports, it is precisely due to the characteristic of intestinal perforation caused by conventional ultrasound endoscopy that the guide wire guidance step is introduced.
Also, I didn't understand what you meant by "follow the water". In the original text of this water injection technology, delivery of EUS mainly relies on ultrasound scanning images, as the endoscopic field of view is indeed limited after water injection. Water should mainly play a role in expanding the intestinal loop and preventing perforation.
I have made modifications to all the other questions you raised. Thank you again for your correction!

Reviewer 2 Report

Comments and Suggestions for Authors Dear Authors, I read the manuscript with interest. This review describes a very interesting and under-researched field of EUS. The paper deserves to be published in Diagnostics after corrections. I would have the following suggestions for authors: 1. Line 36 "...istography" use an adequate term 2. Lines 47 and 48 "...occlusal biopsy and drill biopsy" use adequate terms 3. Line 53 Explain the abbreviation TCB 4. Line 114-116 reformulate the sentence, what is meant by "without assistance of foreign objects" 5. Chapter 2.2. Technical safety needs to be restructured, it is quite difficult to monitor 6. In the text of the manuscript, it is necessary to adequately cite citations from the literature. 7. Authors of cited works should be listed only by last name, not first name and last name (e.g. Nakano et al.) 8. References should be cited in accordance with the provisions of the journal    

Comments on the Quality of English Language

It is necessary to carry out English editing again and correct numerous grammatical and typographical errors

Author Response

Thank you very much for your valuable feedback! I have revised my English and expanded the references. In addition, I have correlated the content of the complications section and adjusted the order of the article content to make its logic more concise and understandable.

Thank you again for your suggestions!

Round 2

Reviewer 1 Report

Comments and Suggestions for Authors

 Congratulations, now the work much better.